# Prevalence and factors associated with inappropriate anti- diabetic medication therapy among type 2 diabetes mellitus patients at the medical and surgical wards of Mbarara Regional Referral Hospital, Uganda

Konjit Abebe Nigussie[1,2]*, Efrata Ashuro Shegena[1], Obwoya Paul Stephen[3], Juliet Sanyu Namugambe[1], Tadele Mekuriya Yadesa[1,2,4]

1 Department of Pharmacy, Mbarara University of Science and Technology, Mbarara, Uganda, 2 PHARMBIOTRAC, World Bank's ACE-II Project, Mbarara University of Science and Technology, Mbarara, Uganda, 3 Department of Internal Medicine, Mbarara University of Science and Technology, Mbarara, Uganda, 4 Department of Pharmacy, College of Medicine & Health Sciences, Ambo University, Ambo, Ethiopia

* koni.connection@gmail.com

## Abstract

### Background

Inappropriate Anti-diabetic Medication Therapy (IADT) refers to a drug-related problem and includes 'ineffective drug therapy', 'unnecessary drug therapy', 'dosage too high', and 'dosage too low'. This study aimed to determine the prevalence and factors associated with IADT among T2DM patients at Mbarara Regional Referral Hospital, Uganda (MRRH).

### Method

A prospective cross-sectional study was conducted at the medical and surgical wards of MRRH from November 2021 to January 2022. One hundred and thirty-eight adult patients aged 18 years and above, with T2DM, were recruited using consecutive sampling. Patient file reviews and interviewer-administered questionnaire was used for data collection. The data were entered into and analyzed using SPSS version 25. Descriptive analysis was employed to describe the population and determine the prevalence of IADT. Types of IADTs were identified using Cipolle's DRP classification tool. A univariate and multivariate logistic regression analysis was used to identify factors significantly associated with IADT. The P-value of < 0.05 was considered statistically significant at 95% confidence interval.

### Results

A total of 138 hospitalized T2DM patients were studied. Eighty (58.0%) were females, and 70 (50.7%) were ≥ 60 years of age. Out of a total of 138 participants, 97 experienced at least one IADT, with an estimated prevalence of 70.3%. 'Dosage too high' (29.2%) and 'dosage too low' (27.9%) were the most common type of IADTs. Age ≥ 60 years (AOR, 8.44;

**Data Availability Statement:** All relevant data are within the manuscript, all data needed to replicate all figures, tables, statistics, and other values are provided.

**Funding:** This work was funded by Pharm-Bio Technology and Traditional Medicine Center (PHARMBIOTRAC). The funders had no role in study design, data collection and analysis, decision to publish, or preparation of the manuscript.

**Competing interests:** The authors have declared that no competing interests exist.

**Abbreviations:** ADA, American Diabetes Association; CAD, Coronary Artery Disease; COVID-19, Coronavirus Disease; CKD, Chronic Kidney Disease; CVD, Cardiovascular Disease; DKD, Diabetic Kidney Disease; DM, Diabetes mellitus; DVT, Deep Vein Thrombosis; DN, Diabetic Nephropathy; DR, Diabetic Retinopathy; DRPs, Drug Related Problems; FBS, Fasting Blood Sugar; TB, Tuberculosis; BPH, Benign Prostatic Hyperplasia; HBA1C, Hemoglobin A1c; AKI, Acute Kidney Injury; COPD, Chronic Obstructive Pulmonary Disease; IADT, Inappropriate Ant-Diabetic Therapy; IDF, International Diabetes Federation; HIV, Human Immunodeficiency Virus; KSA, Kingdom of Saudi Arabia; SGLT2, Sodium Glucose Cotransporter-2; DPP4, Dipeptidyl Peptidase-4 inhibitor; MOH, Ministry of Health; FOM-FRC, Faculty of Medicine Faculty Research Committee; MRRH, Mbarara Regional Refferal Hospital; NCDs, Non-communicable Diseases; PAD, Peripheral Arterial Disease; PCNE, Pharmaceutical Care Network of Europe; T2DM, Type two diabetes mellitus; MUST-REC, Mbarara University of Science and Technology Research Ethics Committee; UGX, Ugandan Shilling; UCG, Ugandan Clinical Guideline; UDA, Ugandan Diabetes Association; WHO, World Health Organization.

95% CI, 2.09–10.90; P-value = 0.003), T2DM duration of < 1 year (AOR, 0.37; 95% CI, 0.11–0.35; P-value = 0.019), and HbA1c of < 7% (AOR, 9.97; 95% CI, 2.34–13.57; P-value = 0.002) were found to be factors significantly associated with the occurrence of IADTs.

## Conclusion

The overall prevalence of inappropriate anti-diabetic medication therapy among T2DM patients admitted to medical and surgical wards of MRRH was 70.3%. The most common type of IADT in this study was 'dosage too high', accounting for almost one-third followed by 'dosage too low' accounting for a quarter of total IADTs. Age greater or equal to 60 years, T2DM duration of < 1 year, and HbA1c of < 7% during the current admission were found to be factors significantly associated with the occurrence of IADTs in hospitalized T2DM patients.

## Introduction

Diabetes mellitus (DM) is a metabolic disease characterized by the state of hyperglycemia as a result of a change in insulin secretion and action [1]. Growing urbanization, sedentary lifestyle, consumption of high caloric food, and stressful lifestyle have led to an increase in the prevalence of DM [2]. DM is classified as type 1, type 2, gestational, and specific types of diabetes due to other causes [3], whereas type 2 diabetes (T2DM) accounts for 90–95% of all diabetes [4]. Pathogenesis of T2DM and development of insulin resistance are characterized by multistimuli factors notably glucolipotoxicity, generation of reactive oxygen species, epigenetic factors, activation of various transcriptional mediated pathways along with the augmented levels of various pro-inflammatory cytokines (Fig 1). Among the various pro-inflammatory cytokines, tumor necrosis factor-alpha (TNF-α) is one of the most important pro-inflammatory mediators that is critically involved in the development of insulin resistance and pathogenesis of T2DM [5]. Exposure to aflatoxin M1 and bisphenol A is also among the causative factors that potentiate the several risk factors notably inflammatory responses and oxidative stress that ultimately induce the pathogenesis of T2DM and associated metabolic disorders [6, 7]. Further, obesity, hypertension, and smoking are observed to be the most critical risk factors accompanying Pro/Ala mutation in peroxisome proliferator-activated receptor gamma (PPAR-γ), thus are associated with a high risk of DM [8]. DM is one of the most common and rapidly growing chronic non-communicable diseases (NCD), with a worldwide prevalence of 6.4% that is estimated to increase to 7.7% by the year 2030 [9]. In Sub-Saharan Africa, diabetes prevalence has grown greatly over the years, and the prevalence rate in Tanzania, Nigeria, and Ethiopia was reported as 8.3%, 4.6%, and 3.2% respectively [10–12]. Similarly in Cameroon and urban Kenya, diabetes prevalence accounted for 5.8% and 12% respectively [13, 14].

Drug-related problem (DRP) is any undesirable event experienced by a patient that involves, or is suspected to involve drug therapy, and that interferes with achieving the desired goals of therapy. According to Cipolle DRPs can be categorized into seven types. These are 'needs additional drug therapy', 'unnecessary drug therapy', 'dose too low', 'ineffective drug therapy', 'dose too high', 'adverse drug reaction', and 'non-compliance' [15]. DRP is a worldwide health problem that compromises the quality of life, increases hospitalization; increases overall health care cost and mortality [16]. Globally, many studies have shown DRPs to be very common in primary care and hospital settings and that a substantial proportion of DRPs occurs among patients with diabetes [17].

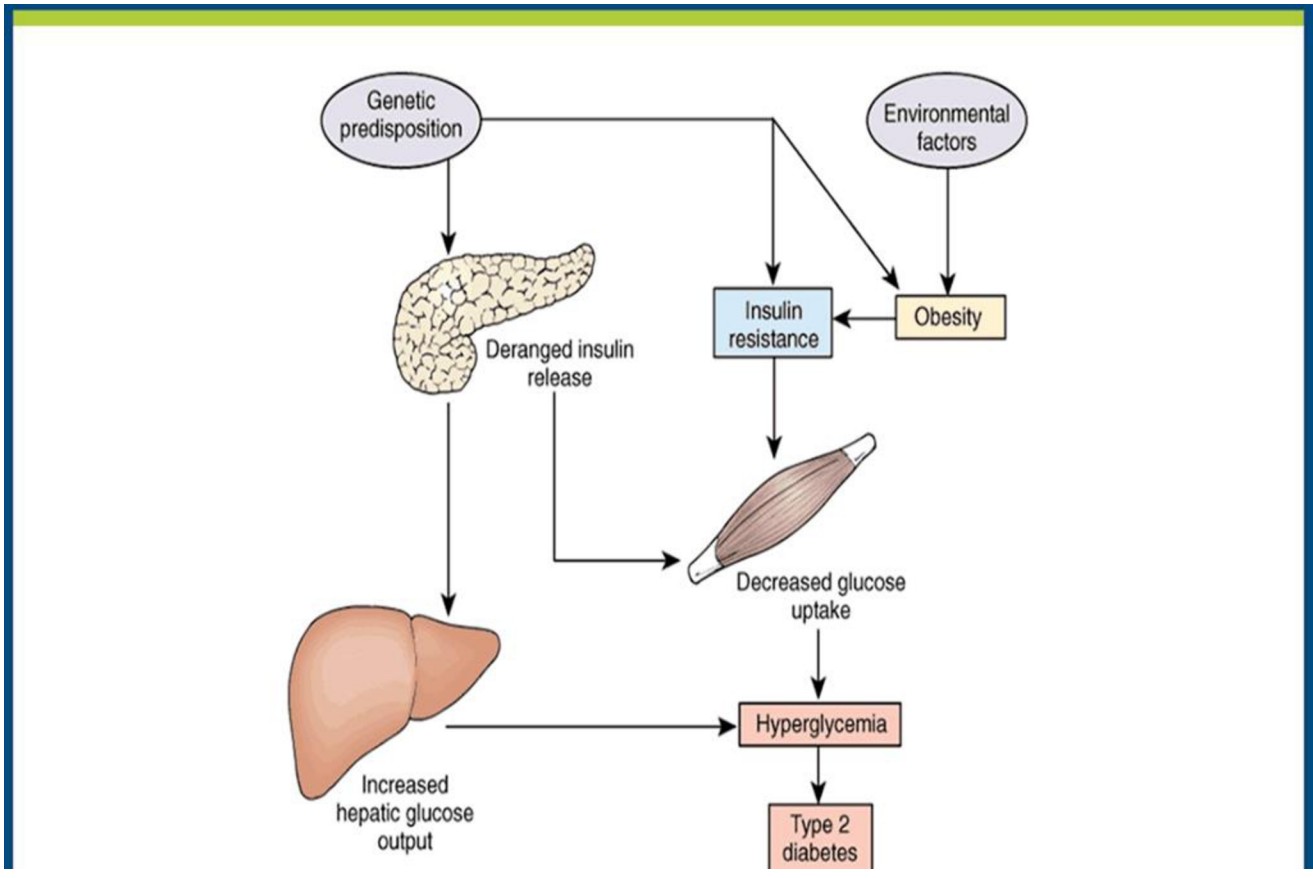

**Fig 1. Pathogenesis of type 2 diabetes mellitus [42].**

Studies worldwide indicate that more than half of diabetic patients on treatment suffer from DRP [18]. A study conducted among T2DM patients in a similar setting in India, Jordan, South Australia, Malaysia, and Ethiopia indicated the prevalence of DRP to be within the range of 71% -91.8% [19–23]. In Kenya, a study among inpatient T2DM patients reported the prevalence of DRP to be 91.1% [24], whereas in Nigeria prevalence of DRP was 2.1% per patient [25]. Another study in Harar, Ethiopia, revealed 'dosage too low' to be the most common type of DRP encountered (36.2%), followed by 'unnecessary drug therapy' (19.7%) and 'ineffective drug therapy' (19.7%) [26]. The occurrence of a DRP among diabetic patients could prevent or delay patients from achieving desired therapeutic goals; contributing to poor glycemic control, and leading to poorer treatment outcomes [27]. Uncontrolled diabetes due to failure of proper use of anti-diabetic medications can lead to increased morbidity and mortality in patients with T2DM [28, 29]. Inappropriate anti-diabetic medication therapy (IADT) severely compromises the effectiveness of treatment making this a critical issue in population health both from the perspective of quality of life and health economics [30]. Inappropriate anti-diabetic medication therapy (IADT) refers to a drug-related problem that is not most suitable for the condition and includes 'ineffective drug therapy', 'unnecessary drug therapy', 'dosage too high', and 'dosage too low'.

Many associated factors have been identified to contribute to the development of DRPs in T2DM patients [25]. These factors include; the presence of comorbidity, poly-pharmacy, age, duration of diabetes, and gender [25, 31–34]. Identifying contributing factors of IADT makes

it possible for better control of diabetes and associated complications and can lead to a significant reduction in morbidity, mortality, and health care costs [35, 36]. Ugandan Ministry of Health revealed that 400,000 Ugandans are diabetic, representing 1.4% of the population (MOH, 2020 [41]). In rural Uganda, the prevalence of T2DM ranged from 1% to 12% [37, 38]. This situation challenges the country already heavily burdened with communicable diseases, financial constraints, and limited medical resources for diabetes care [39]. To the best of our knowledge, there is a limited published study from Uganda on inappropriate anti-diabetic medication therapy. Therefore, the present study assessed the prevalence and factors associated with inappropriate anti-diabetic medication therapy among T2DM patients in the medical and surgical wards of Mbarara Regional Referral Hospital.

## Materials and methods

### Study design and period

A hospital-based prospective cross-sectional study was conducted among inpatient T2DM patients at the medical and surgical wards of Mbarara Regional Referral Hospital from November 2021 to January 2022.

### Study setting

The study was conducted at Mbarara Regional Referral Hospital in Mbarara District, 260 km from Kampala, Uganda. The hospital was founded in 1940 and it is a government-owned referral hospital in the southwestern region and serves some catchment districts and counties. It is also a teaching hospital for Mbarara University of Science and Technology, currently training doctors, nurses, pharmacists, laboratory technologists, and other allied health professionals. The hospital also offers a wide range of health services in the departments of pediatrics, obstetrics and gynecology, internal medicine, surgery, cancer unit, emergency and critical care, imaging, pathology, laboratories, and outpatient department. The diabetes clinic runs once a week under the Internal Medicine department. It attracts approximately 150 to 200 patients per week as recorded in the medical report book. The medical ward is one of the busiest wards in the hospital with a bed capacity of 45 and attracts about 40 patients with diabetes per month, whereas the surgical ward attracts about 10 patients per month with a bed capacity of 60.

### Study population

All adult T2DM patients admitted to the medical and surgical wards of MRRH during the period of study and fulfilled the eligibility criteria.

### Eligibility criteria

**Inclusion criteria.**  All adult patients with T2DM admitted to the medical and surgical ward of MRRH, 18 years of age and above, Patients who received at least one anti-diabetic drug, and Patients who consented to participate during the scheduled study period.

**Exclusion criteria.**  Type 1 DM patients and Patients who were unable to respond to the interview questions (critically ill patients) were excluded from the study.

### Sampling techniques

**Sample size.**  The sample size was calculated using single proportion formula, with a prevalence of 91.1% based on the previous study done in Kenya at Kenyatta National Hospital [24] with the addition of 10% contingency for nonresponse rate.

The Cochran formula was used to determine the sample size as follows [40].

$$n = Z^2\ \frac{p(1-p)}{e^2}$$

Where:

Z- Level of significance is 1.96%

p- Prevalence of DRPs in T2DM is 91.1%

e- Precision estimate around DRPs in T2DM is 5% (0.05)

n- A sample size of DRPs in T2DM.

Assuming 91.1% of the prevalence of DRPs among these patients, the sample size was:

n = 1.962*0.911*(1–0.911) / (0.05)2 = 124.59~125

The sample size was adjusted to 10% non-response. Therefore the number of participants that were recruited for the study was 137.5 ~138.

**Sampling methods.** A consecutive sampling method was used to collect data. Patients found during the study period in the study area who fulfilled the inclusion criteria were sampled.

**Study variables.** The outcome or dependent variable was inappropriate anti-diabetic medication therapy.

The independent variables were:

1. Patient factors include age, gender, alcohol, smoking, marital and education status

2. Disease factors include duration of diabetes, comorbidities, and the number of comorbidities.

3. Drug factors include the number of drugs, class of anti-diabetic drugs, concurrent medications

## Research instrument and data collection

Patients who met the inclusion criteria were approached and explained the purpose of the study and a consent form was completed for patients who agreed to participate before collecting information from them. Once the consent was obtained, a structured questionnaire was used to collect patients' demographics. A data abstraction form was used to collect data on the disease characteristic of patients with T2DM and medication characteristics such as anti-diabetics used, concurrent medication, and a total number of drugs per patient. Each documented drug therapy was evaluated to obtain information on drug appropriateness, recommended dosages, frequency, and duration, based on UCG guideline [41], and ADA guideline [42] was used to obtain some information that was not addressed by UCG. Patients were interviewed for important information missing on their files and were followed until discharge from the respective wards. The data collection was carried out by the principal investigator and one trained research assistant, who was a clinical pharmacy resident at MRRH. The average fasting blood sugar was calculated from three consecutive values to determine the level of glycemic control. DRPs were classified and assessed based on Cipolle et al., 2012 classification [15]. The classification tool has been validated and was used in many other published studies to assess DRP Occurrence [22–24, 26, 32, 43]. The method was refined based on literature review and standard treatment guidelines with further revision, and endorsement by a panel of experts and Clinical Pharmacy Specialists. During data collection all research assistants, study participants, and the principal investigator adhered to the COVID-19 prevention protocol.

**Data management.**   Confirmation of completion of the questionnaires took place after each interview by the principal investigator and any missing information was clarified by a particular participant. After data collection, the data was entered and stored in SPSS and for the confidentiality of participant's information, all files and directories were protected by a password.

**Data analysis.**   Statistical analysis was conducted using Statistical Package for Social Sciences (SPSS) version 25.0 (Armonk, NY: IBM Corp.). Descriptive statistics like frequency, mean, and percentage were generated for continuous and categorical variables and categorical data was expressed as percentages and continuous data was expressed as mean and standard deviation.

The prevalence of inappropriate anti-diabetic medication therapy was calculated by dividing the total number of T2DM patients with at least one IADT by the total number of patients studied and expressed as a percentage. The association between patient, drug, and disease characteristics with IADT was tested using univariate and multivariate logistic regression. Variables with P-value < 0.25 during the univariate analysis were transferred to multivariate analysis. A P-value < 0.05 was considered statistically significant at a 95% confidence interval (CI).

**Quality assurance.**   To control the data quality, training was given to the data collector and the data abstraction form was pre-tested involving 10 patients from both the surgical and medical wards for consistency of understanding of the review tools and completeness of data items. The necessary adjustments were made to the final data extraction format and the filled formats were checked daily by the principal investigator. Also, questionnaires for collecting patient demographic data were pre-tested several times on participants who weren't included in the data collected. To avoid the risk of picking on the same participant in the subsequent sessions, files of patients who had already participated in the study were marked and all information gathered was documented, processed, and presented anonymously and no individual identifiers were collected on the data collection forms, thus no one could trace back the patient's identity. Data were double-checked before entry by the principal investigator.

## Ethical consideration

Ethical approval was obtained from the MUST Faculty of Medicine Faculty Research Committee (FOM-FRC) and the Research Ethics Committee of Mbarara University of Science and Technology (MUST-REC). Site clearance was obtained from the Hospital Director of MRRH. Furthermore, written informed consent was obtained from the participants as per the Ethical Committee guidelines, after explaining to them in English or Runyankole. The consent procedure was approved by the Faculty Research Ethics Committee of MUST. Signed copies of the consent participation forms were kept in a locker that only the principal and research assistants can access them. During the data collection and analysis process, study serial numbers were generated and were used instead of patient names and contact details. For purposes of privacy, patients were interviewed after the general ward round has ended on a one-on-one basis for demographic information and information missing on their medication chart. The collected information was stored on password-protected computers, and accessed only by the principal investigator.

## Results

### Socio-demographic characteristics of the study participants

A total of 138 patients from the medical and surgical wards of MRRH were studied. Eighty (58.0%) were females and 70 (50.7%) were ≥ 60 years of age. The mean age of the study participants was 58±16.1 years. Eighty-two (59.4%) were married, 76 (55.1%) had a primary level of

**Table 1. Sociodemographic characteristics of T2DM patients at medical and surgical wards of MRRH from November 2021 to January 2022, Mbarara, Uganda.**

| Variables | Characteristics | Frequency | Percentage (%) |
|---|---|---|---|
| Gender | Male | 58 | 42.0 |
| | Female | 80 | 58.0 |
| Age | <60 | 68 | 49.3 |
| (Mean ± SD = 58 ± 16.1) | ≥60 | 70 | 50.7 |
| Marital status | Single | 13 | 9.4 |
| | Married | 82 | 59.4 |
| | Separated | 43 | 31.2 |
| Smoking status | Smoker | 68 | 49.3 |
| Alcohol use | Yes | 92 | 66.7 |
| Residence | Rural | 84 | 60.9 |
| | Urban | 54 | 39.1 |
| Employment status | Unemployed | 36 | 26.1 |
| | Self employed | 79 | 57.2 |
| | Employed | 23 | 16.7 |
| Level of education | Illiterate | 24 | 17.4 |
| | Primary | 76 | 55.1 |
| | Secondary | 14 | 10.1 |
| | Tertiary | 24 | 17.4 |
| Religion | Christian | 102 | 73.9 |
| | Muslims | 36 | 26.1 |
| Monthly income | <240000 | 117 | 84.8 |
| | > = 240000 | 21 | 15.2 |

SD: Standard deviation.

education, 84 (60.9%) resided in rural areas, 79(57.2%) were self-employed, and 117(84.8%) reported a monthly income of <240,000ugx (Table 1).

## Disease-related and medication-related characteristics

Most (125, 90.6%) of the participants had the previous history of hospitalization, 102 (73.9%) had a family history of diabetes mellitus, and 135 (97.8%) reported at least one comorbid condition, of which 81(58.7%) patients had ≥ 2 comorbidities. Forty-three (31.2%) participants had a T2DM duration of > 10 years. The average fasting blood sugar was greater than 130 mg/ dl in 58.0% and HbA1c was greater than or equal to 7% in 55.9% of participants respectively. At least one diabetes complication was reported by 54 (39.1%) participants (Table 2).

The commonly prescribed class of anti-diabetic was insulin (89, 64.5%), followed by biguanides (86, 62.3%). Monotherapy with insulin was the frequently prescribed anti-diabetic regimen (51, 37.0%), followed by a combination of Insulin with metformin (32, 23.2%). Nine (6.5%) participants had clinically significant drug-drug interaction and 72 (52.2%) had reported herbal medicine use. At least one concurrent medication use was reported by 132 (95.7%) patients (Table 3 and Fig 2).

## Prevalence of Inappropriate Anti-diabetic Medication Therapy (IADT)

Out of a total of 138 participants, 97 experienced at least one IADT, with an estimated prevalence of 70.3% (95% CI, 62.3–77.5%). Each patient experienced an average of 1.7 ± 0.46 IADTs. Fifty nine (42.8%) participants had 1 IADT, 34 (24.6%) had 2 IADT and 5 (3.6%) participants had 3 IADT (Fig 3).

**Table 2. Disease related factors among T2DM patients at medical and surgical wards of MRRH from November 2021 to January 2022, Mbarara, Uganda.**

| Variable | | Frequency | Percentage (%) |
|---|---|---|---|
| **Family history of DM** | | 102 | 73.9 |
| **Previous history of hospitalization** | | 125 | 90.6 |
| **FBS (mg/dl)** | < 70 | 11 | 8.0 |
| | 70–130 | 47 | 34.1 |
| | > 130 | 80 | 58.0 |
| **HbA1c** | < 7% | 45 | 44.1 |
| | ≥ 7% | 57 | 55.9 |
| **Duration of DM** | < 1 year | 54 | 39.1 |
| | 1–10 years | 41 | 29.7 |
| | > 10 years | 43 | 31.2 |
| **T2DM complications** | At least one | 54 | 39.1 |
| | Nephropathy | 24 | 17.1 |
| | Foot ulcer | 19 | 13.8 |
| | Neuropathy | 15 | 10.9 |
| | PAD | 7 | 5.1 |
| | Retinopathy | 3 | 2.2 |
| **Comorbidity** | At least one | 135 | 97.8 |
| | Hypertension | 78 | 56.5 |
| | CKD | 30 | 21.7 |
| | Peptic ulcer | 30 | 21.7 |
| | HIV | 19 | 13.8 |
| | Heart Failure | 12 | 8.7 |
| | Epilepsy | 9 | 6.5 |
| | Stroke | 8 | 5.7 |
| | Asthma | 8 | 5.7 |
| | Hyperlipidemia | 7 | 5.1 |
| | Liver disease | 6 | 4.3 |
| | Stomach cancer | 5 | 3.6 |
| | Others* | 31 | 22.5 |
| **No of comorbidities** | 0–1 | 57 | 41.3 |
| | > = 2 | 81 | 58.7 |

*Schizophrenia, Prostate cancer, AKI, COPD, Pancreatic cancer, TB, BPH, DVT, Crohn's disease.

## Types of inappropriate anti-diabetic medication therapy

Out of a total of 247 IADTs, 'Dosage too high' (72, 29.2%) accounted for the highest proportion followed by 'dosage too low' (69, 27.9%) (Fig 4).

## Factors associated with inappropriate anti-diabetic medication therapy

**Univariate analysis.** Among the factors analyzed for possible association with IADT, age ≥ 60 years (COR, 2.63; 95% CI, 1.23–5.63; P-value = 0.013), history of smoking, family history of DM, T2DM duration of < 1 year (COR, 0.33; 95% CI, 0.13–0.82; P-value = 0.017), HbA1c < 7% (COR, 4.09; 95% CI, 1.49–11.23; P-value = 0.006), T2DM complication, biguanides class of anti-diabetics, sulfonylurea class of anti-diabetics, and DPP4-inhibitor class of anti-diabetics qualified for multivariate logistic regression analysis at P-value of < 0.25 (Table 4).

**Table 3. Medication related factors among T2DM patients at medical and surgical wards of MRRH from November 2021 to January 2022, Mbarara, Uganda.**

| Variable | | Frequency | Percentsge (%) |
|---|---|---|---|
| **Class of anti-diabetics** | Insulin | 89 | 64.5 |
| | Biguanides | 86 | 2.3 |
| | Sulphonylurea | 45 | 32.6 |
| | SGLT2-inhibitors | 9 | 6.5 |
| | DPP4-inhibitors | 7 | 5.1 |
| **Concurrent medication** | At least one | 132 | 95.7 |
| | Cardiovascular agents | 130 | 94.2 |
| | GI agents | 22 | 15.9 |
| | Antivirals | 21 | 15.2 |
| | Antibacterials | 15 | 10.8 |
| | Anticonvulsants | 14 | 10.1 |
| | Respiratory tract agents | 12 | 8.7 |
| | Blood product/modifiers | 9 | 6.5 |
| | Others** | 28 | 20.3 |
| **Clinically significant drug interaction** | | 9 | 6.5 |
| **Herbal medicine use** | | 72 | 52.2 |
| **Total number of drugs per patient** | < 5 medications | 58 | 42.0 |
| | ≥ 5 medications | 80 | 58.0 |

**Analgesics, Antiparastic, Antipsychotics, Antidepressants, Hormonal agents.

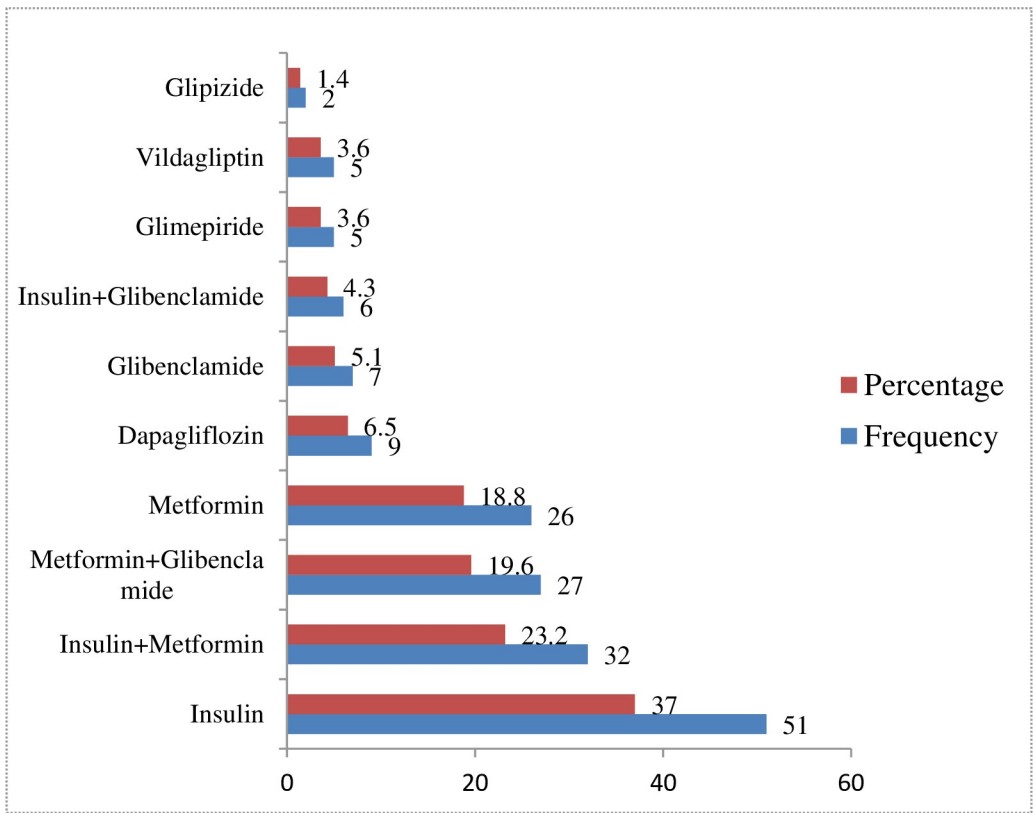

**Fig 2. Anti-diabetics used among T2DM patients at medical and surgical wards of MRRH, Mbarara, Uganda.**

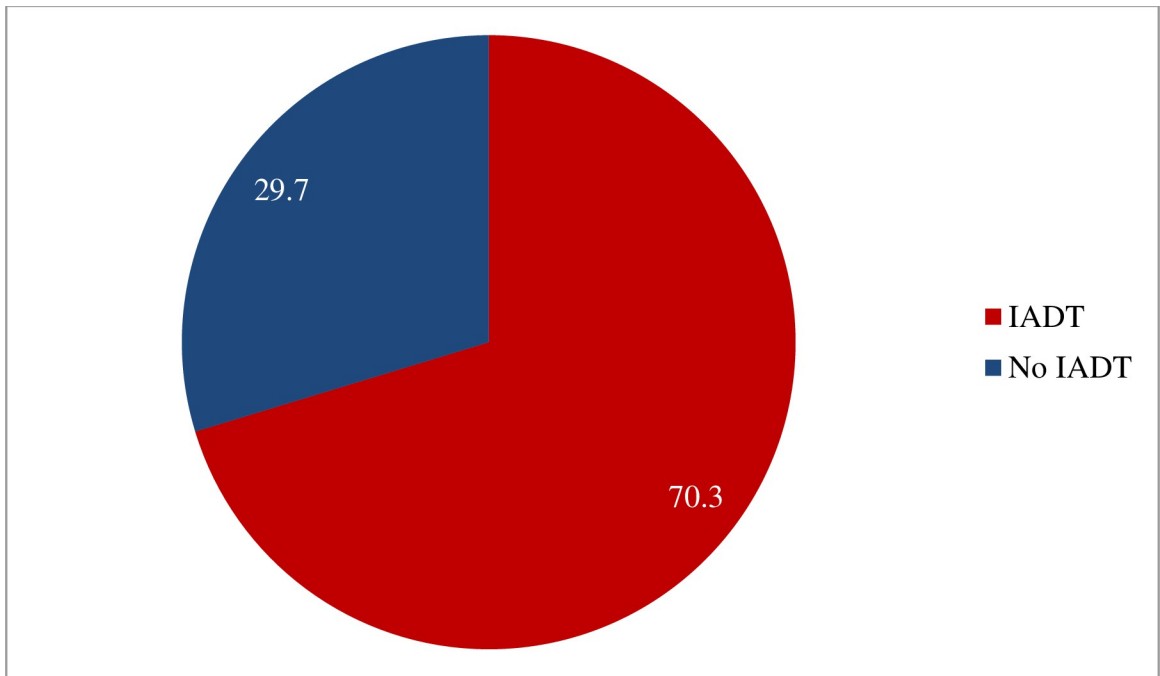

**Fig 3. Prevalence of inappropriate anti-diabetic medication therapy among T2DM patients at medical and surgical wards of MRRH, Mbarara, Uganda.**

**Multivariate analysis.**   Out of the nine independent variables included in multivariate logistic regression, three variables including age $\geq$ 60 years, T2DM duration of $<$ 1 year, and HbA1c of $<$ 7% were determined to factor significantly associated with IADT (Table 4). Patients who were 60 years and older were 8.44 times (AOR, 8.44; 95% CI, 2.09–10.90; P-value = 0.003) more likely to have IADT when compared to those less than 60 years. Patients who had T2DM for less than 1 year were 63% less likely (AOR, 0.37; 95% CI, 0.11–0.35; P-value = 0.019) to have IADT compared to patients with T2DM duration longer than 10 years. Similarly, Patients with glycated hemoglobin of less than 7% had 9.97 times (AOR, 9.97; 95% CI, 2.34–13.57; P-value = 0.002) higher odds of experiencing IADT compared to patients with HbA1c of 7% and higher.

## Discussion

The current study showed that 70.3% of hospitalized T2DM patients had at least one inappropriate anti-diabetic medication therapy with an average of 1.7 ± 0.46 IADTs. This is consistent with the prevalence's of DRPs in patients with T2DM of 70% in the US [21], 71.9% in Ethiopia [44], 71.1% in India [45], and 73.7% in Pakistan [34]. On the other hand, the prevalence of DRPs in this study is lower than in the DRP studies in Malaysia (91.8% and 90.5%) [20, 46], Nigeria (94%), Australia (96%), and Switzerland (100%) [25, 47, 48]. The discrepancy with the previous studies could be due to the use of different study populations, sample sizes, and study designs; the studies in Malaysia and Switzerland involved T2DM patients with dyslipidemia and hypertension comorbidity, had a large sample size and used retrospective study design. The discrepancies that exist could also be attributed to the difference in the study protocol and participants who had taken four or more drug therapy were the inclusion criteria in the above three studies: those patients receiving complex drug regimens are vulnerable to the existence of medication-related problems and would have more DRPs. In addition, previous studies

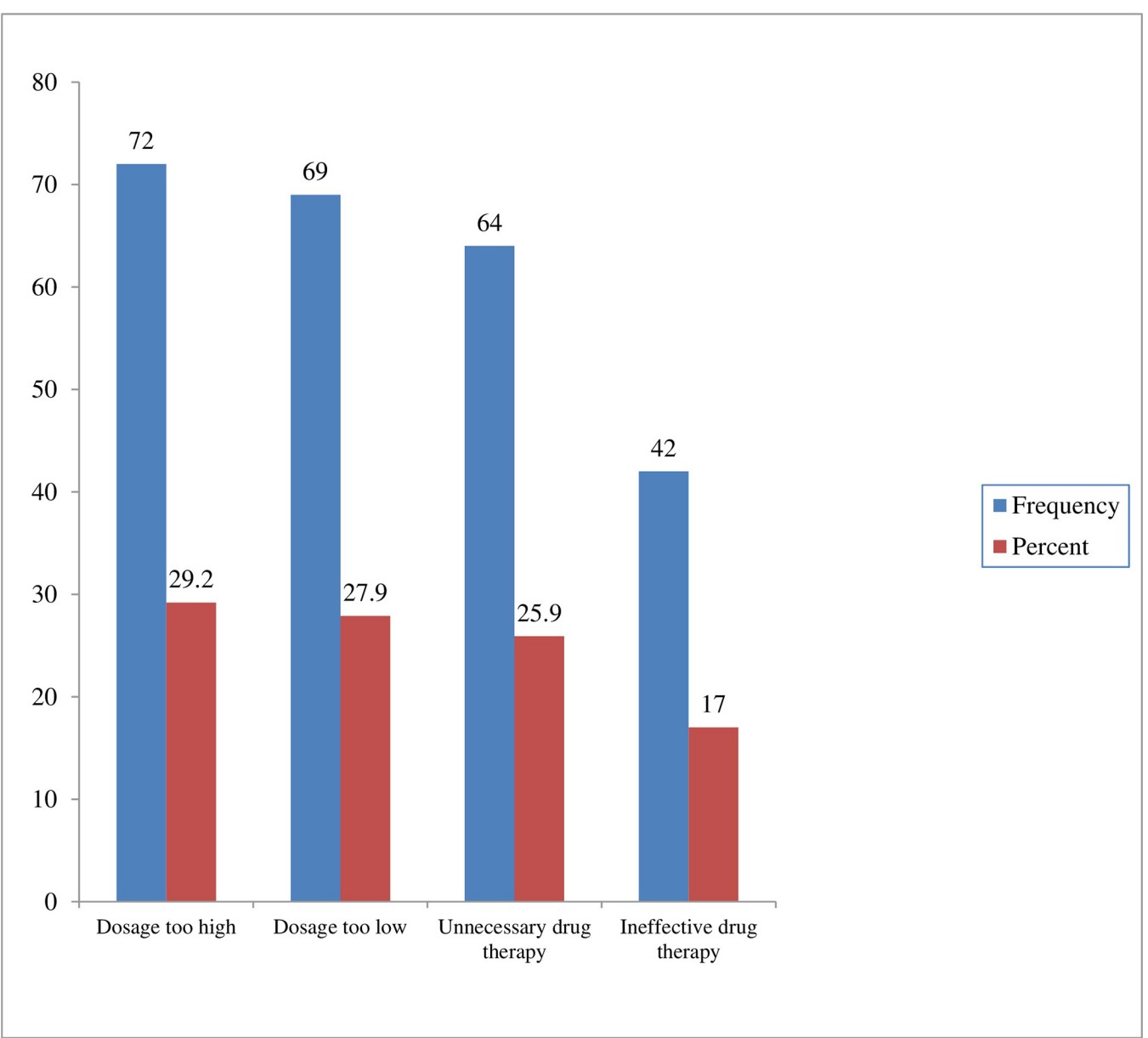

**Fig 4. Types of inappropriate anti-diabetic medication therapy among T2DM patients at medical and surgical wards of MRRH, Mbarara, Uganda.**

used the PCNE classification of DRPs. On the contrary, our study used Cipolle's DRP classification tool with the exclusion of three types of DRPs and identified DRPs involving only anti-diabetic medications, which might have contributed to the low prevalence of DRP in our setting. Our study highlights the need to prioritize the burden of DRPs for designing an effective intervention to mitigate problems related to drug therapy and underscored the need to conduct further research on the study site.

The most common type of IADT identified in this study was 'Dosage too high' (29.2%). This is in line with a study in Malaysia (27.9%) [20]. This proportion is higher than three studies which reported 1.6 to 21.6% [24, 26, 47]. This difference might be attributed to a lack of assessment of patients' renal and hepatic functions and failure to consider patients' age and weight when prescribing anti-diabetic medications at the study site. Furthermore, the high

**Table 4. Logistic regression analysis of the association between independent variables and IADT among T2DM patients at medical and surgical wards of MRRH from November 2021 to January 2022, Mbarara, Uganda.**

| Variables | | IADT | | COR (95% CI) | P-value | AOR (95% CI) | P-value |
|---|---|---|---|---|---|---|---|
| | | No | Yes | | | | |
| **Gender** | Male | 18 | 40 | 0.80 (0.43–1.88) | 0.772 | | |
| | Female | 23 | 57 | 1 | | | |
| **Age***** | < 60 years | 27 | 41 | 1 | | 1 | |
| | ≥ 60 years | 14 | 56 | 2.63(1.23–5.63) | 0.013 | 8.44 (2.09–10.90) | 0.003# |
| **FBS (mg/dl)** | <70 | 2 | 9 | 1 | | | |
| | 70–130 | 16 | 31 | 1.82 (0.36–9.05) | 0.467 | | |
| | > 130 | 23 | 57 | 0.78 (0.36–1.60) | 0.533 | | |
| **Marital status** | Single | 4 | 9 | 1 | | | |
| | Married | 22 | 60 | 1.21 (0.32–4.58) | 0.784 | | |
| | Separated | 15 | 28 | 1.46 (0.66–3.24) | 0.350 | | |
| **Smoking***** | Smoker | 15 | 53 | 2.09 (0.99–4.42) | 0.055 | 2.86 (0.83–9.78) | 0.095 |
| | Non-smoker | 26 | 44 | 1 | | 1 | |
| **Alcohol use** | Yes | 25 | 67 | 1.43 (0.67–3.06) | 0.358 | | |
| | No | 16 | 30 | 1 | | | |
| **Residence area** | Rural | 22 | 62 | 1 | | 1 | |
| | Urban | 19 | 35 | 1.53 (0.73–3.21) | 0.261 | | |
| **Employement status** | Unemployed | 13 | 23 | 1 | | | |
| | Self-employed | 22 | 57 | 0.62 0.(197–1.98) | 0.423 | | |
| | Employed | 6 | 17 | 0.91 (0.32–2.62) | 0.868 | | |
| **Religion** | Christian | 30 | 72 | 1.06 (0.46–2.42) | 0.897 | | |
| | Muslim | 11 | 25 | 1 | | | |
| **Monthly income (UGX)** | < 240,000 | 36 | 81 | 1 | | | |
| | ≥ 240,000 | 5 | 16 | 0.70 (0.24–2.07) | 0.522 | | |
| **Family history of DM***** | Yes | 27 | 75 | 1.77 (0.79–3.94) | 0.164 | 1.047 (0.28–3.99) | 0.946 |
| | No | 14 | 22 | 1 | | 1 | |
| **Previous history of hospitalization** | Yes | 36 | 89 | 1.55 (0.47–5.04) | 0.471 | | |
| | No | 5 | 8 | 1 | | | |
| **Duration of T2DM***** | < 1 year | 24 | 30 | 0.33 (0.13–0.82) | 0.017 | 0.37(0.11–0.35) | 0.019# |
| | 1–10 year | 8 | 33 | 1.09 (0.38–3.17) | 0.872 | 2.15 (0.49–9.44) | 0.309 |
| | > 10 year | 9 | 34 | 1 | | 1 | |
| **HbA1c***** | < 7% | 6 | 39 | 4.09(1.49–11.23) | 0.006 | 9.97 (2.34–13.57) | 0.002# |
| | ≥ 7% | 22 | 35 | 1 | | 1 | |
| **Number of comorbidity** | 0–1 | 14 | 43 | 1 | | | |
| | ≥ 2 | 27 | 54 | 0.65 (0.31–1.39) | 0.268 | | |
| **T2DM complication***** | Yes | 21 | 33 | 0.49 (0.23–1.03) | 0.061 | 0.73(0.22–2.45) | 0.606 |
| | No | 20 | 64 | 1 | | 1 | |
| **Class of anti-diabetics*** (Biguanides)** | Yes | 22 | 64 | 1.68 (0.79–3.52) | 0.174 | 0.22 (0.020–2.46) | 0.22 |
| | No | 19 | 33 | 1 | | 1 | |
| **Class of anti-diabetics*** (Sulphonylureas)** | Yes | 7 | 38 | 3.128 (1.26–7.77) | 0.014 | 1.611 (0.39–6.64) | 0.509 |
| | No | 34 | 59 | 1 | | 1 | |
| **Class of anti-diabetics (Insulin)** | Yes | 28 | 61 | 0.79 (0.36–1.71) | 0.545 | | |
| | No | 13 | 36 | 1 | | | |
| **DPP-4 inhibitor***** | Yes | 4 | 3 | 0.20 (0.06–1.38) | 0.122 | 0.57 (0.06–5.30) | 0.619 |
| | No | 37 | 94 | 1 | | 1 | |

(*Continued*)

**Table 4.** (Continued)

| Variables | | IADT | | COR (95% CI) | P-value | AOR (95% CI) | P-value |
|---|---|---|---|---|---|---|---|
| | | No | Yes | | | | |
| Class of anti-diabetics (SGLT2-inhibitor) | Yes | 3 | 6 | 0.84 (0.199–3.51) | 0.806 | | |
| | No | 38 | 91 | 1 | | | |
| Drug interaction | Yes | 3 | 6 | 0.84 (0.199–3.51) | 0.806 | | |
| | No | 38 | 91 | 1 | | | |
| Herbal medicine use | Yes | 20 | 52 | 1.21 (0.58–2.52) | 0.604 | | |
| | No | 21 | 45 | 1 | | | |
| Concurrent medicine use | Yes | 40 | 92 | 0.46 (0.05–4.07) | 0.485 | | |
| | No | 1 | 5 | 1 | | | |
| Total number of drugs | < 5 medications | 15 | 43 | 1 | | | |
| | ≥ 5 medications | 26 | 54 | 1.38 0.65–2.93 | 0.400 | | |

***Variables that reached multivariate regression

#Statically signifcant at P-value < 0.05, CI = confidence interval, COR = crude odds ratio, AOR = adjusted odds ratio.

occurrence of 'dosage too high' in our study is probably due to a lack of compliance to standard diabetes treatment guidelines, too frequent dosage regimen, drug-drug interaction, and inter-patient variation in drug response which was beyond the scope of the present study. This puts patients at higher risk of dosage-related adverse drug reactions. Dose adjustment of drugs in patients with impaired renal function and low estimated glomerular filtration rate may prevent adverse effects and decrease unnecessary drug expenditures [49, 50].

The second most common type of IADT in this study was 'dosage too low' accounting for 27.9% of the total IADTs. This is comparable with a study in Ethiopia (28.0%) and another study in Kenya (24.9%) [24, 51]. On the other hand, this proportion is higher than five studies which reported 1.3 to 15.8% [33, 45, 46, 52, 53]. Such a high occurrence of 'dosage too low' in our study could be attributed to incorrect storage of insulin-based regimens, ineffective dose or dosage regimen not frequent enough, and drug-drug interactions. In addition, the high proportion of 'dosage too low' may also be due to the underuse of standard diabetes treatment guidelines, and dose calculation for insulin-based regimens was majorly based on assumed weight. This explains the failure to attain the desired glycemic control among our study participants. The current study showed that more than half of the participants had poor glycemic control. Therefore, efforts should be made to minimize drug dosing problems by improving the adherence to standard treatment guidelines, increasing the involvement of clinical pharmacists in deciding the dosing of drugs, or the implementation of computerized dosing program [50].

The proportion of 'unnecessary drug therapy' in our study was 25.9%, which is in agreement with a study in Eastern Ethiopia (26.7%) [26]. However, this proportion is higher than a study in Nigeria and two studies in Ethiopia, ranging from 3.6 to 17.6% [23, 32, 54]. This high proportion in our study site may be due to the underuse of standard diabetes treatment guidelines, initiation of insulin-based regimens without clear indications, and initiation of anti-diabetics when non-drug therapy is more appropriate. 'Ineffective drug therapy' which accounted for 17.0% was the least common IADT identified in our study and was attributed to the use of the ineffective drugs when a 'more effective drug was available'. However, this proportion is lower than a study in Indonesia and Nigeria, which reported 50% and 29.6% respectively (Kisno et al., 2011) [54]. This discrepancy was due to non-adherence to guidelines and inter-patient variation in drug response in the former studies.

The multivariate logistic regression showed that age greater than or equal to 60 years, T2DM duration of < 1 year, and HbA1c of < 7% were significantly associated with the occurrence of IADTs. It is found that patients greater or equal to 60 years of age were 8.44 times more likely (AOR, 8.44; 95% CI, 2.09–10.90; P-value = 0.003) to experience IADT than patients in the age group of < 60 years. This finding is in agreement with three studies in Ethiopia [23, 33, 55]. Many studies have concluded that elderly patients with multi-morbidity and taking multiple medications were more likely to experience drug-drug interactions, and thus more likely to develop DRPs, particularly dosages too high and dosages too low [56–59]. Moreover, age-related physiologic changes influence both pharmacokinetics and pharmacodynamics of drugs in elderly patients. The slowing of metabolism and excretory process with aging increases the chance of problems arising from drugs, especially dosages too high [60, 61]. Hence, special attention is needed to prevent the occurrence of DRPs in these patients.

Additionally, participants who had a T2DM duration of less than 1 year were 63% less likely to have IADT compared to patients with a T2DM duration of more than 10 years (AOR, 0.37; 95% CI, 0.11–0.35; P-value = 0.019). This finding is similar to a study in Ethiopia [31], India [62], and Pakistan [34]. This might be because those patients with a longer duration of diabetes had a higher chance of developing diabetes complications which contribute to drug-drug interactions and multiple drug therapy. In addition, these patients are likely to have comorbid conditions which influence the desired outcome of other diseases by increasing the number of drugs, causing drug-drug, drug-disease, and disease-disease interactions which collectively resulted in increased likelihood of experiencing DRP in the study patients [31–34]. It was also found that patients with HbA1c of less than 7% during the current admission had 9.97 times higher (AOR, 9.97; 95% CI, 2.34–13.57; P-value = 0.002) odds of experiencing IADTs compared to patients with HbA1c of greater or equal to 7%. To achieve HbA1c of < 7% patients are more likely to be initiated on multiple anti-hyperglycemic agents and this increases the occurrence of IADTs especially 'dosage too high' and 'unnecessary drug therapy'. Furthermore, aggressive pharmaceutical therapy increases both side effects and costs. Hence, healthcare professionals should avoid aggressive goals of HbA1c in those at high risk of DRPs including the elderly.

## Study limitation

The prevalence of IADT was compared with the prevalence of DRP in other studies as no study specifically focused on IADT. In addition, this study was conducted in only one study center. Hence, further studies, which consider these variables will be needed to solve these limitations.

## Conclusion

The overall prevalence of inappropriate anti-diabetic medication therapy among T2DM patients admitted to the medical and surgical wards of MRRH was 70.3%. The most common type of IADT in this study was 'dosage too high', accounting for almost one-third followed by 'dosage too low' accounting for a quarter of total IADTs. Age greater or equal to 60 years, T2DM duration of < 1 year, and HbA1c of < 7% during the current admission were found to be factors significantly associated with the occurrence of IADTs in hospitalized T2DM patients.

## Acknowledgments

The authors are sincerely grateful to the patients and caregivers that offered to participate in this study and most importantly, the authors are thankful for the support and approval granted

by the administration of Mbarara Regional Referral Hospital (MRRH) for allowing us to conduct the study.

## Author Contributions

**Conceptualization:** Konjit Abebe Nigussie, Efrata Ashuro Shegena, Obwoya Paul Stephen, Juliet Sanyu Namugambe, Tadele Mekuriya Yadesa.

**Data curation:** Konjit Abebe Nigussie, Efrata Ashuro Shegena, Obwoya Paul Stephen, Juliet Sanyu Namugambe, Tadele Mekuriya Yadesa.

**Formal analysis:** Konjit Abebe Nigussie, Efrata Ashuro Shegena, Obwoya Paul Stephen, Juliet Sanyu Namugambe, Tadele Mekuriya Yadesa.

**Methodology:** Konjit Abebe Nigussie, Efrata Ashuro Shegena, Obwoya Paul Stephen, Juliet Sanyu Namugambe, Tadele Mekuriya Yadesa.

**Project administration:** Konjit Abebe Nigussie, Efrata Ashuro Shegena, Obwoya Paul Stephen, Juliet Sanyu Namugambe, Tadele Mekuriya Yadesa.

**Resources:** Konjit Abebe Nigussie.

**Software:** Konjit Abebe Nigussie, Efrata Ashuro Shegena, Tadele Mekuriya Yadesa.

**Supervision:** Obwoya Paul Stephen, Juliet Sanyu Namugambe, Tadele Mekuriya Yadesa.

**Validation:** Konjit Abebe Nigussie, Efrata Ashuro Shegena, Obwoya Paul Stephen, Juliet Sanyu Namugambe, Tadele Mekuriya Yadesa.

**Writing – original draft:** Konjit Abebe Nigussie, Efrata Ashuro Shegena, Obwoya Paul Stephen, Juliet Sanyu Namugambe, Tadele Mekuriya Yadesa.

**Writing – review & editing:** Konjit Abebe Nigussie, Efrata Ashuro Shegena, Obwoya Paul Stephen, Juliet Sanyu Namugambe, Tadele Mekuriya Yadesa.

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
