## [Decision Letter · Decision Letter 0]

1 May 2022

PONE-D-22-08952Prevalence and Factors Associated with Inappropriate Anti- diabetic Medication Therapy among Type 2 Diabetes Mellitus Patients at Medical and Surgical Wards of Mbarara Regional Referral Hospital, Uganda.PLOS ONE

Dear Dr. Nigussie,

Thank you for submitting your manuscript to PLOS ONE. After careful consideration, we feel that it has merit but does not fully meet PLOS ONE’s publication criteria as it currently stands. Therefore, we invite you to submit a revised version of the manuscript that addresses the points raised during the review process.

I have received the reports from our advisors on your manuscript which you submitted to PLOS ONE.

Based on the comments received, I feel that your manuscript could be reconsidered for publication should you be prepared to incorporate major revisions.

When preparing your revised manuscript, you are asked to carefully consider the reviewer comments below and submit a list of responses to the comments.

Editor Comments: There is a huge list of grammatical mistakes and syntax errors. The paper should be checked by a professional speaker of English before complete acceptance.

We look forward to receiving your revised manuscript.

Kind regards,

Muhammad Sajid Hamid Akash

Academic Editor

PLOS ONE

Journal Requirements:

a) Did participants provide their written or verbal informed consent to participate in this study?

"The authors thank PHARMBIOTRAC, Mbarara University of Science and Technology, a project of World Bank ACE-II for supporting the corresponding author’s tuition for her Msc

studies and for funding this research work."

"This work was funded by Pharm-Bio Technology and Traditional Medicine Center (PHARMBIOTRAC).

5. Please ensure that you refer to Figure 3 in your text as, if accepted, production will need this reference to link the reader to the figure.

6. Please upload a copy of Figure 5, to which you refer in your text on page 13. If the figure is no longer to be included as part of the submission please remove all reference to it within the text.

7. We note you have included a table to which you do not refer in the text of your manuscript. Please ensure that you refer to Table 4 in your text; if accepted, production will need this reference to link the reader to the Table.

8. Please include a copy of Table 5 which you refer to in your text on page 15.

Reviewers' comments:

Reviewer's Responses to Questions

**Comments to the Author**

1. Is the manuscript technically sound, and do the data support the conclusions?

Reviewer #1: No

2. Has the statistical analysis been performed appropriately and rigorously? 

Reviewer #1: No

3. Have the authors made all data underlying the findings in their manuscript fully available?

Reviewer #1: No

4. Is the manuscript presented in an intelligible fashion and written in standard English?

Reviewer #1: No

5. Review Comments to the Author

Reviewer #1: The author presented an original article on ‘Prevalence and Factors Associated with Inappropriate Anti- diabetic Medication Therapy among Type 2 Diabetes Mellitus Patients at Medical and Surgical Wards of Mbarara Regional Referral Hospital, Uganda’. The manuscript needs to be improved regarding the following aspects.

• There is no association between key points and conclusion.

• Backgrounds of Diabetes are not discussed. Add biomarkers for diabetes and various risk factors like oxidative stress, retinopathy, neuropathy, etc. Review these articles Environ Sci Pollut Res. 2021; 28(33): 44818–44832. https://doi.org/10.1007/s11356-021-14871-w; J Pak Med Assoc. 2021; 71(1): 286-96. https://doi.org/10.47391/JPMA.434; Rev Endocr Metab Disord. 2020; 21(4): 631-43. https://doi.org/10.1007/s11154-020-09549-6; Environ Sci Pollut Res. 2020; 27(21): 26262-26275. https://doi.org/10.1007/s11356-020-09044-0; Clin Exp Pharmacol Physiol. 2020; 47(9):1517–1529. https://doi.org/10.1111/1440-1681.13339; Adv Exp Med Biol. 2019;1084:95-107 https://doi.org/10.1007/5584_2018_195; J Cell Biochem. 2018;119(1):105-10. https://doi.org/10.1002/jcb.26174; J Cell Biochem. 2017;118(11):3577-85. https://doi.org/10.1002/jcb.26097; Crit Rev Eukaryot Gene Expr. 2016;26(4): 317-32. https://doi.org/10.1615/CritRevEukaryotGeneExpr.2016016782; J Cell Biochem. 2013;114(3):525-31. https://doi.org/10.1002/jcb.24402; Curr Diabetes Rev. 2013;9(5):387-96. https://doi.org/10.2174/15733998113099990069; Curr Diabetes Rev. 2013;9(4):286-93. https://doi.org/10.2174/15733998113099990062; Eds; Al-Gubory KH, Laher I. Springer-Verlag (Germany). 2018: 377-395. https://doi.org/10.1007/978-3-319-67625-8_15.It is important to highlight the pathogenesis and risk factors associated with diabetes.

• Future prospective of study is not mentioned.

• Results and discussion are not appropriate.

• Abbreviations are missing.

• Literature is not up to date needs to be reviewed properly.

• Diagrams and figures are missing .Add figures and diagrams as well showing pathogenesis of diabetes.

• There are many formatting mistakes.

• There are some grammatical errors in this manuscript, such as verbs and prepositions. The manuscript needs extensive review by an author.

6. PLOS authors have the option to publish the peer review history of their article (what does this mean?). If published, this will include your full peer review and any attached files.

Reviewer #1: **Yes: **Momina Shahid

---

## [Author Response · Author response to Decision Letter 0]

3 Jun 2022

I have responded to the specific comments raised by the reviewers and attached it as a separate file titled 'Response to Reviewers'.

---

## [Editor Report · Decision Letter 1]

6 Jun 2022

Prevalence and Factors Associated with Inappropriate Anti- diabetic Medication Therapy among Type 2 Diabetes Mellitus Patients at the Medical and Surgical Wards of Mbarara Regional Referral Hospital, Uganda.

PONE-D-22-08952R1

Dear Dr. Nigussie,

We’re pleased to inform you that your manuscript has been judged scientifically suitable for publication and will be formally accepted for publication once it meets all outstanding technical requirements.

Kind regards,

Muhammad Sajid Hamid Akash

Academic Editor

PLOS ONE
---

## [Editor Report · Acceptance letter]

17 Jun 2022

PONE-D-22-08952R1 

Prevalence and Factors Associated with Inappropriate Anti- diabetic Medication Therapy among Type 2 Diabetes Mellitus Patients at the Medical and Surgical Wards of Mbarara Regional Referral Hospital, Uganda. 

Dear Dr. Nigussie:

I'm pleased to inform you that your manuscript has been deemed suitable for publication in PLOS ONE. Congratulations! Your manuscript is now with our production department. 

Kind regards, 

on behalf of

Dr. Muhammad Sajid Hamid Akash 

Academic Editor

PLOS ONE